# Mirror Descent Methods with Weighting Scheme for Outputs for Constrained Variational Inequality Problems

**Mohammad S. Alkousa**
`m.alkousa@innopolis.ru`
Innopolis University
Innopolis, Universitetskaya Str., 1, 420500, Russia.
Moscow Institute of Physics and Technology
9 Institutsky lane, Dolgoprudny, 141701, Russia.

**Belal A. Alashqar**
`alashkar.ba@phystech.edu`
Moscow Institute of Physics and Technology
9 Institutsky lane, Dolgoprudny, 141701, Russia.

**Fedor S. Stonyakin**
`fedyor@mail.ru`
Moscow Institute of Physics and Technology
9 Institutsky lane, Dolgoprudny, 141701, Russia.
V. I. Vernadsky Crimean Federal University
4 Academician Vernadsky Avenue, Simferopol,
295007, Republic of Crimea, Russia.
Innopolis University
Innopolis, Universitetskaya Str., 1, 420500, Russia.

**Tarek Nabhani**
`tnabhani@su.edu.sa`
Faculty of Science and Humanities
Shaqra University
Al-Dawadmi, Saudi Arabia.

**Seydamet S. Ablaev**
`seydamet.ablaev@yandex.ru`
V. I. Vernadsky Crimean Federal University
4 Academician Vernadsky Avenue, Simferopol,
295007, Republic of Crimea, Russia.

## Abstract

Variational inequalities play a key role in machine learning research such as generative adversarial networks, supervised/unsupervised learning, reinforcement learning, adversarial training, and generative models. This paper is devoted to the variational inequality problems. We consider two classes of problems, the first is classical constrained variational inequality and the second is the same problem with functional (inequality type) constraints. To solve these problems, we propose mirror descent-type methods with a weighting scheme for the generated points in each iteration of the algorithms. This scheme assigns smaller weights to the initial points and larger weights to the most recent points, thus it improves the convergence rate of the proposed methods. For the variational inequality problem with functional constraints, the proposed method switches between adaptive and non-adaptive steps depending on the values of the functional constraints at iterations. We analyze the proposed methods for the time-varying step sizes and prove the optimal convergence rate for variational inequality problems with bounded and monotone operators. The results of numerical experiments of the proposed methods for classical constrained variational inequality problems show a significant improvement over the modified projection method.

## 1 Introduction

Variational inequalities (VIs) cover as a special case many optimization problems such as minimization problems, saddle point problems, and fixed point problems (see Examples 3.1, 3.2 and 3.3, below). They often arise in various mathematical problems, such as optimal control, partial differen-

tial equations, mechanics, finance, and many others. They play a key role in solving equilibrium and complementarity problems Facchinei (2003); Harker & Pang (1990), in machine learning research such as generative adversarial networks Goodfellow et al. (2020), supervised/unsupervised learning Joachims (2005); Xu et al. (2004), reinforcement learning Jin & Sidford (2020); Omidshafiei et al. (2017), adversarial training Madry et al. (2017), and generative models Daskalakis et al. (2017); Gidel et al. (2018).

Numerous researchers have dedicated their efforts to exploring theoretical aspects related to the existence and stability of solutions and constructing iterative methods to solve the classical VIs (by classical, we mean the problems without functional "inequality types" constraints), see equation 7. A significant contribution to developing numerical methods for solving classical VIs was made in the 1970s, when the extragradient method was proposed in Korpelevich (1976). More recently, Nemirovski in his seminal work Nemirovski (2004) proposed a non-Euclidean variant of this method, called the Mirror Prox algorithm, which can be applied to Lipschitz continuous operators. Different methods with similar complexity were also proposed in Auslender & Teboulle (2005); Gasnikov et al. (2019); Nesterov (2007). Besides that, in Nesterov (2007), Nesterov proposed a method for variational inequalities with a bounded variation of the operator, i.e., with a non-smooth operator. There is also extensive literature on variations of the extragradient method that avoid taking two steps or two gradient computations per iteration, and so on (see for example Hsieh et al. (2019); Malitsky & Tam (2020)).

Another important class of VIs is the problem with functional constraints (inequality type), see equation 22. The presence of such type of constraints makes these problems more difficult to solve. This class of problems arises in many fields of mathematics, among them are economic equilibrium models Levin et al. (1993), game theory Garcia & Zangwill (1981), constrained Markov potential games Alatur et al. (2023); Jordan et al. (2024), generalized Nash equilibrium problems with jointly-convex constraints Facchinei & Kanzow (2010); Jordan et al. (2023) and hierarchical programming problems Migdalas & Pardalos (1996), in mathematical physics Baiocchi (1984). See Antipin (2000) to see some examples. In addition, this class of problems encompasses important applications in machine learning, including reinforcement learning with safety constraints Xu et al. (2021) and learning with fairness constraints Lowy et al. (2021); Zafar et al. (2019).

For VIs with functional constraints, the previous works have focused on primal-dual algorithms based on the (augmented) Lagrangian function to handle the constraints and penalty methods Auslender (1999; 2003); He et al. (2004); Zhu (2003). These algorithms and their convergence guarantees crucially depend on information about the optimal Lagrange multipliers. In Zhang et al. (2024), a primal method was proposed without knowing any information on the optimal Lagrange multipliers, proving its convergence rate for the problem with monotone operators under smooth constraints. In Yang et al. (2022), a first order method (ACVI) was presented which combines path-following interior point methods and primal-dual methods. In Chavdarova et al. (2024), the authors proposed a primal-dual approach to solving VIs with general functional constraints by taking the last iteration of ACVI. Although there are many works for the VIs with functional constraints, they remain very few compared to the existing works for the classical constrained problem.

In this paper, we propose Mirror Descent type methods for solving the classical variational inequality problem, and the same problem with functional constraints (inequality types); see problems equation 7 and equation 22. The mirror descent method, for minimization problems, which originated in Nemirovski (1979); Nemirovskij & Yudin (1983) and was later analyzed in Beck & Teboulle (2003), is considered as the non-Euclidean extension of standard subgradient methods. This method is used in many applications, see Ben-Tal et al. (2001); Nazin & Miller (2011); Nazin et al. (2014); Tremba & Nazin (2013) and references therein. The standard subgradient methods employ the Euclidean distance function with a suitable step-size in the projection step. Mirror descent extends standard projected subgradient methods by employing a nonlinear distance function with an optimal step-size in the nonlinear projection step Luong et al. (2016). The Mirror Descent method not only generalizes the standard subgradient methods but also achieves a better convergence rate Doan et al. (2018). It is also applicable to optimization problems in Banach spaces where gradient descent is not Doan et al. (2018). An extension of the mirror descent method for constrained problems was proposed in Beck et al. (2010); Nemirovskij & Yudin (1983). The class of non-smooth optimization problems with non-smooth functional constraints attracts widespread interest in many areas of modern large-scale optimization and its applications Ben-Tal & Nemirovski (1997); Nesterov & Shpirko (2014). In terms of continuous optimization with functional constraints, there is a long history of studies.

The monographs in this area include Bertsekas (2014); Nocedal & Wright (1999). Some of the works on first-order methods for convex optimization problems with convex functional constraints include (for example, but not limited to) Bayandina et al. (2018); Lin et al. (2018); Stonyakin et al. (2018; 2019); Titov et al. (2018) for the deterministic setting and Alkousa (2020; 2019); Lan & Zhou (2020); Xu (2020) for the stochastic setting.

Recently in Zhu et al. (2024), for the projected subgradient method, the optimal convergence rate was proved using the previously mentioned time-varying step size with a new weighting scheme for the generated points in each iteration of the algorithm. This convergence rate remains the same (optimal) even if we slightly increase the weight of the most recent points, thereby relaxing the ergodic sense. These results were recently extended to mirror descent methods for constrained minimization problems in Alkousa et al. (2024b) and for minimization problems with functional constraints (inequality type) in Alkousa et al. (2024a).

In this paper, for the classical constrained variational inequality problem, we propose a mirror descent-type method (Algorithm 1) with a weighting scheme for the points generated in each iteration of the algorithm. We extend Algorithm 1 (see Algorithm 2), to be applicable to the variational inequality problem with functional constraints by switching between adaptive and non-adaptive steps. We analyze the proposed methods for the time-varying step sizes and obtain the optimal convergence rate (for Algorithm 1) for the class of variational inequality problems with bounded and monotone operators.

The paper consists of an introduction and five main sections. In Sect. 2 we mentioned the basic facts, definitions, and tools for variational inequalities. Sect. 3 devoted to the classical constraint variational inequality problem. We proposed a mirror descent method (Algorithm 1) with a weighting scheme for the points generated in each iteration of the algorithm, we analyzed Algorithm 1 and proved its optimal convergence rate for the class of variational inequality problems with bounded and monotone operators. In Sect. 4, we proposed an extension of Algorithm 1 (see 2) to solve a more complicated variational inequality problem with functional constraints. In Sect. 5 we present numerical experiments that demonstrate the efficiency of the proposed weighting scheme in Algorithm 1, and compare its work with a modified projection method, proposed in Khanh & Vuong (2014), to solve some examples of the variational inequality problem. In the last section 6, we review the results obtained in the paper.

## 2 PRELIMINARIES

Let $(\mathbf{E}, \|\cdot\|)$ be a normed finite-dimensional vector space, with an arbitrary norm $\|\cdot\|$, and $\mathbf{E}^*$ be the conjugate space of $\mathbf{E}$ with the following norm

$$\|y\|_* = \max_{x \in \mathbf{E}}\{\langle y, x\rangle : \|x\| \leq 1\},$$

where $\langle y, x\rangle$ is the value of the continuous linear functional $y \in \mathbf{E}^*$ at $x \in \mathbf{E}$.

Let $Q \subset \mathbf{E}^n$ be a compact convex set with diameter $D > 0$, that is, $\max_{x,y \in Q} \|x - y\| \leq D$, and $\psi : Q \longrightarrow \mathbb{R}$ be a proper closed differentiable and $\sigma$-strongly convex (called prox-function or distance generating function). The corresponding Bregman divergence is defined as

$$V_\psi(x, y) = \psi(x) - \psi(y) - \langle \nabla\psi(y), x - y\rangle \quad \forall x, y \in Q.$$

For the Bregmann divergence, it holds the following inequality

$$V_\psi(x, y) \geq \frac{\sigma}{2}\|y - x\|^2 \quad \forall x, y \in Q. \tag{1}$$

**Definition 2.1.** *($\delta$-monotone operator). Let $\delta > 0$. The operator $F : Q \longrightarrow \mathbf{E}^*$ is called $\delta$-monotone, if it holds*

$$\langle F(y) - F(x), y - x\rangle \geq -\delta \quad \forall x, y \in Q. \tag{2}$$

For example, we can consider $F = \nabla_\delta f$ for $\delta$-subgradient $\nabla_\delta f(x)$ of convex function $f$ at a point $x \in Q$: $f(y) - f(x) \geq \langle \nabla_\delta f(x), y - x\rangle - \delta$ for each $y \in Q$ (see e.g., Chapter 5 in Polyak (1987)).

When $\delta = 0$, then the operator $F$ is called monotone, i.e.,

$$\langle F(x) - F(y), x - y\rangle \geq 0 \quad \forall x, y \in Q. \tag{3}$$

We say that the operator $F$ is bounded on $Q$, if there exist $L_F > 0$ such that

$$\|F(x)\|_* \leq L_F \quad \forall x \in Q. \tag{4}$$

The following identity, known as the three points identity, is essential in analyzing the mirror descent method.

**Lemma 2.2.** *(Three points identity Chen & Teboulle (1993)) Suppose that $\psi : \mathbf{E} \longrightarrow (-\infty, \infty]$ is a proper closed, convex, and differentiable function on $\mathrm{dom}(\partial \psi)$. Let $a, b \in \mathrm{dom}(\partial(\psi))$ and $c \in \mathrm{dom}(\psi)$. Then it holds*

$$\langle \nabla \psi(b) - \nabla \psi(a), c - a \rangle = V_\psi(c, a) + V_\psi(a, b) - V_\psi(c, b). \tag{5}$$

**Fenchel-Young inequality**(Beck & Teboulle (2003)). For any $b \in \mathbf{E}, a \in \mathbf{E}^*$, it holds the following inequality

$$\langle a, b \rangle \leq \frac{\|a\|_*^2}{2\lambda} + \frac{\lambda \|b\|^2}{2} \quad \forall \lambda > 0. \tag{6}$$

## 3 MIRROR DESCENT METHOD FOR CONSTRAINED VARIATIONAL INEQUALITY PROBLEM

In this section, we consider the following variational inequality problem

$$\text{Find} \quad x^* \in Q : \quad \langle F(x), x^* - x \rangle \leq 0 \quad \forall x \in Q, \tag{7}$$

where $F : Q \longrightarrow \mathbf{E}^*$ is a continuous, bounded (i.e., equation 4 holds), and $\delta$-monotone operator (i.e., equation 2 holds).

Under the assumption of continuity and monotonicity (i.e., $\delta = 0$) of the operator $F$, the problem equation 7 is equivalent to a Stampacchia Giannessi (1998) (or strong Nesterov (2007)) variational inequality, in which the goal is to find $x^* \in Q$ such that

$$\langle F(x^*), x^* - x \rangle \leq 0 \quad \forall x \in Q. \tag{8}$$

To emphasize the extensiveness of the problem equation 7 (or equation 8), we mention three common special cases for VIs.

**Example 3.1** (Minimization problem). *Let us consider the minimization problem*

$$\min_{x \in Q} f(x), \tag{9}$$

*and assume that $F(x) = \nabla f(x)$, where $\nabla f(x)$ denotes the (sub)gradient of $f$ at $x$. Then, if $f$ is convex, it can be proved that $x^* \in Q$ is a solution to equation 8 if and only if $x^* \in Q$ is a solution to equation 9.*

**Example 3.2** (Saddle point problem). *Let us consider the saddle point problem*

$$\min_{u \in Q_u} \max_{v \in Q_v} f(u, v), \tag{10}$$

*and assume that $F(x) := F(u, v) = (\nabla_u f(u, v), -\nabla_v f(u, v))^\top$, where $Q = Q_u \times Q_v$ with $Q_u \subseteq \mathbb{R}^{n_u}, Q_v \subseteq \mathbb{R}^{n_v}$. Then if $f$ is convex in $u$ and concave in $v$, it can be proved that $x^* \in Q$ is a solution to equation 8 if and only if $x^* = (u^*, v^*) \in Q$ is a solution to equation 10.*

**Example 3.3** (Fixed point problem). *Let us consider the fixed point problem*

$$\text{find} \ x^* \in Q \ \text{such that} \ T(x^*) = x^*, \tag{11}$$

*where $T : \mathbb{R}^n \longrightarrow \mathbb{R}^n$ is an operator. Taking $F(x) = x - T(x)$, it can be proved that $x^* \in Q = \mathbb{R}^n$ is a solution to equation 8 if $F(x^*) = \mathbf{0} \in \mathbb{R}^n$, that is, $x^*$ is a solution to equation 11.*

**Definition 3.4.** *For some $\varepsilon > 0$, we call a point $\widehat{x} \in Q$ an $\varepsilon$-solution of the problem equation 7, if*

$$\langle F(x), \widehat{x} - x \rangle \leq \varepsilon \quad \forall x \in Q. \tag{12}$$

Following Nesterov (2007), to assess the quality of a candidate solution $\widehat{x}$, we use the following restricted gap (or merit) function

$$\text{Gap}(\widehat{x}) = \max_{u \in Q} \langle F(u), \widehat{x} - u \rangle. \tag{13}$$

Thus, our goal is to find an approximate solution to the problem equation 7, that is, a point $\widehat{x} \in Q$ such that the following inequality holds

$$\text{Gap}(\widehat{x}) = \max_{u \in Q} \langle F(u), \widehat{x} - u \rangle \leq \varepsilon, \tag{14}$$

for some $\varepsilon > 0$.

For problem equation 7, we propose an Algorithm 1, under consideration

$$V_\psi(x, y) \leq V_\psi(x, x^1) < \infty \quad \forall x, y \in Q, \tag{15}$$

where $x^1 \in Q$ is a chosen (dependently on $Q$) initial point for the Algorithm 1.

**Remark 3.5.** *In analyzing the proposed algorithms, we are interested in the optimal complexity for the number of iterations to solve the problems with the desired accuracy. Therefore, in the proposed algorithms we assume that the auxiliary minimization problems are simple, in the sense that the set $Q$ does not have a complex structure and the problems can be solved explicitly. Otherwise, for each iteration of the proposed algorithms, we must solve the auxiliary problems numerically and then the analysis will be for the total oracle complexity, which is outside the scope of the research now, which we will address in future work.*

---

**Algorithm 1** Mirror descent method for constrained variational inequality problem.

---

**Require:** step sizes $\{\gamma_k\}_{k \geq 1}$, initial point $x^1 \in Q$ s.t. equation 15 holds, number of iterations $N$.
1: **for** $k = 1, 2, \ldots, N$ **do**
2: $\quad x^{k+1} = \arg\min_{x \in Q} \left\{ \langle x, F(x^k) \rangle + \frac{1}{\gamma_k} V_\psi(x, x^k) \right\}.$
3: **end for**

---

For Algorithm 1, we have the following result.

**Theorem 3.6.** *Let $F : Q \longrightarrow \boldsymbol{E}^*$ be a continuous, bounded, and $\delta$-monotone operator. Then for problem equation 7, by Algorithm 1, with a positive non-increasing sequence of step sizes $\{\gamma_k\}_{k \geq 1}$, for any fixed $m \geq -1$, it satisfies the following inequality*

$$\text{Gap}(\widehat{x}) \leq \frac{1}{\sum_{k=1}^N \gamma_k^{-m}} \left( \frac{R^2}{\gamma_N^{m+1}} + \frac{1}{2\sigma} \sum_{k=1}^N \frac{\|F(x^k)\|_*^2}{\gamma_k^{m-1}} \right) + \delta, \tag{16}$$

*where $R > 0$, such that $\max_{x \in Q} V_\psi(x, x^1) \leq R^2$, and*

$$\widehat{x} = \frac{1}{\sum_{k=1}^N \gamma_k^{-m}} \sum_{k=1}^N \gamma_k^{-m} x^k.$$

Now, let us see the convergence rate of Algorithm 1 with the following (adaptive and non-adaptive) time-varying step size rules

$$\gamma_k = \frac{\sqrt{2\sigma}}{L_F \sqrt{k}}, \quad \text{or} \quad \gamma_k = \frac{\sqrt{2\sigma}}{\|F(x^k)\|_* \sqrt{k}}, \quad k = 1, 2, \ldots, N, \tag{17}$$

and different values of the parameter $m \geq -1$.

**Corollary 3.7.** *Let $F : Q \longrightarrow \boldsymbol{E}^*$ be a continuous, bounded, and $\delta$-monotone operator. Then for problem equation 7, by Algorithm 1, with $m = -1$, and the time-varying step sizes given in equation 17, it satisfies the following*

$$\text{Gap}(\widetilde{x}) \leq \frac{L_F \left( R^2 + 1 + \log(N) \right)}{\sqrt{\sigma}} \cdot \frac{1}{\sqrt{N}} + \delta = O \left( \frac{\log(N)}{\sqrt{N}} \right) + \delta, \tag{18}$$

*where $\widetilde{x} = \frac{1}{\sum_{k=1}^N \gamma_k} \sum_{k=1}^N \gamma_k x^k$.*

Note that the convergence rate in equation 18, is suboptimal for the bounded monotone operators (i.e., when $\delta = 0$ in equation 2).

From Theorem 3.6, with a special value of the parameter $m$, we can obtain the optimal convergence rate $O\left(\frac{1}{\sqrt{N}}\right)$ of Algorithm 1, with the time-varying step sizes given in equation 17 and monotone operators (i.e., $\delta = 0$ in equation 2).

For this, we have the following result.

**Corollary 3.8.** *Let $F : Q \longrightarrow \mathbf{E}^*$ be a continuous, bounded, and $\delta$-monotone operator. Then for problem equation 7, by Algorithm 1, with $m = 0$, and the time-varying step sizes given in equation 17, it satisfies the following*

$$\mathrm{Gap}(\overline{x}) \leq \frac{L_F\left(2 + R^2\right)}{\sqrt{2\sigma}} \cdot \frac{1}{\sqrt{N}} + \delta = O\left(\frac{1}{\sqrt{N}}\right) + \delta, \tag{19}$$

*where $\overline{x} = \frac{1}{N} \sum_{k=1}^{N} x^k$.*

Also, the same optimal convergence rate for Algorithm 1, with bounded monotone operators (i.e., $\delta = 0$), can be obtained with any fixed $m \geq 1$, and time-varying step sizes given in equation 17.

For this, we have the following result.

**Corollary 3.9.** *Let $F : Q \longrightarrow \mathbf{E}^*$ be a continuous, bounded, and $\delta$-monotone operator. Then for problem equation 7, by Algorithm 1, with any $m \geq 1$, and the time-varying step sizes given in equation 17, it satisfies the following*

$$\mathrm{Gap}(\widehat{x}) \leq \frac{L_F(m+2)(1+R^2)}{2\sqrt{2\sigma}} \cdot \frac{1}{\sqrt{N}} + \delta = O\left(\frac{1}{\sqrt{N}}\right) + \delta, \tag{20}$$

*where $\widehat{x} = \frac{1}{\sum_{k=1}^{N} \gamma_k^{-m}} \sum_{k=1}^{N} \gamma_k^{-m} x^k$.*

**Remark 3.10.** *In comparison with the suboptimal convergence rate equation 18, when $m \geq 1$, the weighting scheme $\frac{1}{\sum_{k=1}^{N} \gamma_k^{-m}} \sum_{k=1}^{N} \gamma_k^{-m} x^k$ assigns smaller weights to the initial points and larger weights to the most recent points generated by Algorithm 1. This fact will be shown in numerical experiments (see Sect. 5). In addition, note that the parameter $m$ does not affect the convergence rate (from a theoretical point of view). But in the experiments (see Sect. 5), it has a good effect on the convergence. Where the algorithm significantly gives a better solution as $m$ increases while we do not face the case of division on zero, thus $m$ is bounded above with a constant to avoid a critical experimental situation.*

## 4    MIRROR-DESCENT METHOD FOR VARIATIONAL INEQUALITY PROBLEM WITH FUNCTIONAL CONSTRAINTS

Consider a set of convex subdifferentiable functionals $g_i : Q \longrightarrow \mathbb{R}$, $i = 1, 2, \ldots, p$. Also we assume that all functionals $g_i$ are Lipschitz-continuous with some constant $M_{g_i} > 0$, i.e.,

$$|g_i(x) - g_i(y)| \leq M_{g_i}\|x - y\| \quad \forall\, x, y \in Q \text{ and } i = 1, \ldots, p. \tag{21}$$

This means that at every point $x \in Q$ and for any $i = 1, \ldots, p$ there is a subgradient $\nabla g_i(x)$, such that $\|\nabla g_i(x)\|_* \leq M_{g_i}$.

In this section, we consider the following variational inequality problem

$$\begin{aligned} \text{Find} \quad x^* \in Q : \quad & \langle F(x), x^* - x\rangle \leq 0 \quad \forall x \in Q, \\ & \text{and} \quad g_i(x) \leq 0 \quad \forall i = 1, 2 \ldots, p, \end{aligned} \tag{22}$$

where $F : Q \longrightarrow \mathbf{E}^*$ is a continuous, bounded (i.e., equation 4 holds), and $\delta$-monotone operator (i.e., equation 2 holds).

It is clear that instead of a set of Lipschitz-continuous functions $\{g_i(\cdot)\}_{i=1}^{p}$ we can see one Lipschitz-continuous functional constraint $g : Q \longrightarrow \mathbb{R}$, such that

$$g(x) = \max_{1 \leq i \leq p} \{g_i(x)\}, \text{ and } |g(x) - g(y)| \leq M_g\|x - y\| \quad \forall\, x, y \in Q, \tag{23}$$

where $M_g = \max_{1 \leq i \leq p}\{M_{g_i}\}$. Thus, the problem equation 22, will be equivalent to the following problem

$$\text{Find} \quad x^* \in Q : \quad \langle F(x), x^* - x \rangle \leq 0, \quad \text{and} \quad g(x) \leq 0, \quad \forall x \in Q. \tag{24}$$

**Definition 4.1.** *For some $\varepsilon > 0$, we call a point $\widehat{x} \in Q$ an $\varepsilon$-solution of the problem equation 24, if*

$$\langle F(x), \widehat{x} - x \rangle \leq \varepsilon \quad \forall x \in Q, \quad \text{and} \quad g(\widehat{x}) \leq \varepsilon. \tag{25}$$

To solve the problem equation 22 (or its equivalent equation 24), we propose a mirror-decent type method, listed as Algorithm 2 below.

As can be seen from the items of Algorithm 2, the needed point (i.e., the output, see equation 27) is selected among the points $x^i$ for which $g(x^i) \leq \varepsilon$. Therefore, we will call step $i$ *productive* if $g(x^i) \leq \varepsilon$. If the reverse inequality $g(x^i) > \varepsilon$ holds, then the step $i$ will be called *non-productive*.

Let $I$ and $J$ denote the set of indices of productive and non-productive steps, respectively. $|A|$ denotes the cardinality of the set $A$. Let us also set $\gamma_k := \gamma_k^F$ if $k \in I$, $\gamma_k := \gamma_k^g$ if $k \in J$.

---

**Algorithm 2** Mirror descent algorithm for VIs with functional constraints.

---

**Require:** $\varepsilon > 0$, initial point $x^1 \in Q$, step sizes $\{\gamma_k^F\}_{k \geq 1}$, $\{\gamma_k^g\}_{k \geq 1}$, number of iterations $N$.
1: $I \longrightarrow \emptyset, J \longrightarrow \emptyset$.
2: **for** $k = 1, 2, \ldots, N$ **do**
3:     **if** $g(x^k) \leq \varepsilon$ **then**
4:         $x^{k+1} = \arg\min_{x \in Q}\left\{\langle x, F(x^k)\rangle + \frac{1}{\gamma_k^F} V_\psi(x, x^k)\right\}$.
5:         $k \longrightarrow I$     "productive step"
6:     **else**
7:         Calculate $\nabla g(x^k) \in \partial g(x^k)$,
8:         $x^{k+1} = \arg\min_{x \in Q}\left\{\langle x, \nabla g(x^k)\rangle + \frac{1}{\gamma_k^g} V_\psi(x, x^k)\right\}$.
9:         $k \longrightarrow J$     "non-productive step"
10:     **end if**
11: **end for**

---

For Algorithm 2, we have the following result.

**Theorem 4.2.** *Let $F : Q \longrightarrow E^*$ be a continuous, bounded, and $\delta$-monotone operator. Let $g(x) = \max_{1 \leq i \leq p}\{g_i(x)\}$ be an $M_g$-Lipschitz convex function, where $g_i : Q \longrightarrow \mathbb{R}$, $\forall i = 1, 2, \ldots, p$ are $M_{g_i}$-Lipschitz convex functions, and $M_g = \max_{1 \leq i \leq p}\{M_{g_i}\}$. Then for problem equation 24, by Algorithm 2, with a positive non-increasing sequence of step sizes $\{\gamma_k\}_{k \geq 1}$, for any fixed $m \geq -1$, after $N \geq 1$ iterations, it satisfies the following inequality*

$$\text{Gap}(\widehat{x}) < \frac{1}{\sum_{k \in I}(\gamma_k^F)^{-m}}\left(\frac{R^2}{\gamma_N^{m+1}} + \frac{1}{2\sigma}\sum_{k \in I}\frac{\|F(x^k)\|_*^2}{(\gamma_k^F)^{m-1}} + \frac{1}{2\sigma}\sum_{k \in J}\frac{\|\nabla g(x^k)\|_*^2}{(\gamma_k^g)^{m-1}}\right.$$
$$\left. - (\varepsilon - M_g D)\sum_{k \in J}(\gamma_k^g)^{-m}\right) + \delta, \tag{26}$$

*where $R > 0$, such that $\max_{x \in Q} V_\psi(x, x^1) \leq R^2$, $D > 0$ is the diameter of $Q$, and*

$$\widehat{x} = \frac{1}{\sum_{k \in I}(\gamma_k^F)^{-m}}\sum_{k \in I}(\gamma_k^F)^{-m}x^k. \tag{27}$$

**Remark 4.3** (Stopping rule for Algorithm 2). *From Theorem 4.2, with*

$$\widehat{x} = \frac{1}{\sum_{i \in I}(\gamma_i^F)^{-m}}\sum_{i \in I}(\gamma_i^F)^{-m}x^i,$$

*for any $k \geq 1$, we find*

$$\left(\sum_{i \in I} \left(\gamma_i^F\right)^{-m}\right) \text{Gap}(\widehat{x}) < \frac{R^2}{\gamma_k^{m+1}} + \frac{1}{2\sigma} \sum_{i \in I} \frac{\|F(x^i)\|_*^2}{(\gamma_i^F)^{m-1}} + \frac{1}{2\sigma} \sum_{j \in J} \frac{\|\nabla g(x^j)\|_*^2}{(\gamma_j^g)^{m-1}}$$

$$- (\varepsilon - M_g D) \sum_{i=1}^{k} (\gamma_i)^{-m} + (\varepsilon - M_g D) \sum_{i \in I} (\gamma_i^F)^{-m} + \delta \sum_{i \in I} \left(\gamma_i^F\right)^{-m}$$

$$= (\delta + \varepsilon) \sum_{i \in I} \left(\gamma_i^F\right)^{-m} - \left( (\varepsilon - M_g D) \sum_{i=1}^{k} (\gamma_i)^{-m} + M_g D \sum_{i \in I} \left(\gamma_i^F\right)^{-m}\right.$$

$$\left. - \frac{R^2}{\gamma_k^{m+1}} - \frac{1}{2\sigma} \sum_{i \in I} \frac{\|F(x^i)\|_*^2}{(\gamma_i^F)^{m-1}} - \frac{1}{2\sigma} \sum_{j \in J} \frac{\|\nabla g(x^j)\|_*^2}{(\gamma_j^g)^{m-1}}\right).$$

*From this, without relying on prior knowledge of the number of iterations $N$ that the algorithm performs, we can set for any $k \geq 1$,*

$$M_g D \sum_{i \in I} \left(\gamma_i^F\right)^{-m} \geq \frac{R^2}{\gamma_k^{m+1}} + \frac{1}{2\sigma} \sum_{i \in I} \frac{\|F(x^i)\|_*^2}{(\gamma_i^F)^{m-1}} + \frac{1}{2\sigma} \sum_{j \in J} \frac{\|\nabla g(x^j)\|_*^2}{(\gamma_j^g)^{m-1}} + (M_g D - \varepsilon) \sum_{i=1}^{k} (\gamma_i)^{-m}$$

$$(28)$$

*as a stopping rule of Algorithm 2. As a result, we conclude*

$$\left(\sum_{i \in I} \left(\gamma_i^F\right)^{-m}\right) \max_{x \in Q} \langle F(x), \widehat{x} - x \rangle < (\delta + \varepsilon) \sum_{i \in I} \left(\gamma_i^F\right)^{-m}.$$

*Thus,*

$$\max_{x \in Q} \langle F(x), \widehat{x} - x \rangle < \delta + \varepsilon.$$

*Note that for all $i \in I$ it holds that $g(x^i) \leq \varepsilon$, and since $g$ is convex, then we have*

$$g(\widehat{x}) \leq \frac{1}{\sum_{i \in I} \left(\gamma_i^F\right)^{-m}} \sum_{i \in I} \left(\gamma_i^f\right)^{-m} g(x^i) \leq \varepsilon.$$

*Thus after the stopping rule equation 28 of Algorithm 2 is met we find $\widehat{x}$, which is given in equation 27, such that*

$$\text{Gap}(\widehat{x}) = \max_{x \in Q} \langle F(x), \widehat{x} - x \rangle < \delta + \varepsilon, \quad and \quad g(\widehat{x}) \leq \varepsilon.$$

In Appendix C, we provide an analysis of Algorithm 2 with a variant of time-varying step size rules.

## 5 NUMERICAL EXPERIMENTS

To show the advantages and effects of the considered weighting scheme for generated points by Algorithm 1 (see Theorem 3.6) in its convergence, a series of numerical experiments were performed for some examples of the classical variational inequality problem. We compare the performance of Algorithm 1 with the Modified Projection Method (MPM) proposed in Khanh & Vuong (2014). In our experiments, we take the standard Euclidean prox-structure, namely $\psi(x) = \frac{1}{2}\|x\|_2^2$ which is 1-strongly functions (i.e., $\sigma = 1$) and the corresponding Bregman divergence is $V_\psi(x, y) = \frac{1}{2}\|x - y\|_2^2$. In all experiments, we take the set $Q$ as a unit ball in $\mathbb{R}^n$ with the center at $\mathbf{0} \in \mathbb{R}^n$. The compared methods start from the same initial point $x^1 = \left(\frac{1}{\sqrt{n}}, \ldots, \frac{1}{\sqrt{n}}\right) \in \mathbb{R}^n$. The results of the comparisons for the considered examples are presented in Figs. 1 and 2. These results show the values $\|F_k/F(x^1)\|_2^2$, where $F_k := F(\widehat{x}_k)$, and $\widehat{x}_k = \frac{1}{\sum_{i=1}^{k} \gamma_i^{-m}} \sum_{i=1}^{k} \gamma_i^{-m} x^i$.

**Example 5.1.** *(Dong et al. (2018)) Let $F : \mathbb{R}^2 \longrightarrow \mathbb{R}^2$ be a monotone and bounded operator in the unit ball, defined as follows*

$$F(x_1, x_2) = (2x_1 + 2x_2 + \sin(x_1), -2x_1 + 2x_2 + \sin(x_2)). \tag{29}$$

**Example 5.2.** *(Sahu & Singh (2021)) Let $F : \mathbb{R}^3 \longrightarrow \mathbb{R}^3$ be a monotone and bounded operator in the unit ball, defined as follows*

$$F(x_1, x_2, x_3) = (F_1(x_1, x_2, x_3), F_2(x_1, x_2, x_3), F_3(x_1, x_2, x_3)), \tag{30}$$

*where $r, s, t \in \mathbb{R}$ and $F_1(x_1, x_2, x_3) = x_1 - sx_2 + tx_3 + \sin(x_1), F_2(x_1, x_2, x_3) = x_2 - rx_3 + sx_1 + \sin(x_2), F_3(x_1, x_2, x_3) = x_3 - tx_1 + rx_2 + \sin(x_3).$*

The results for Examples 5.1 and 5.2, presented in Fig. 1. From this figure, we can see that MPM works better than Algorithm 1 only for small values of the parameter $m$. But Algorithm 1 works better than MPM, with a big difference between their performance when we increase the value of the parameter $m$.

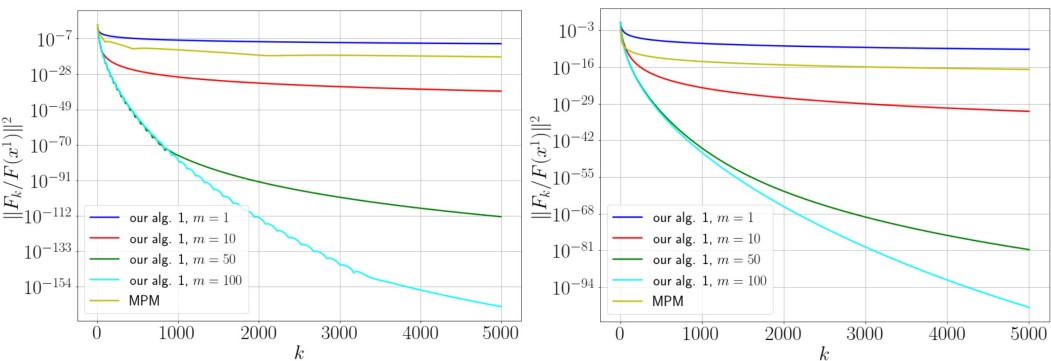

**Figure 1:** Results of Algorithm 1 and Modified Projection Method Khanh & Vuong (2014), for Example 5.1 (in the left) and for Example 5.2 (in the right).

**Example 5.3.** *In this example, we consider the HpHard (or Harker-Pang) problem Qu et al. (2024). This problem is a well-known issue in nonnegative matrix factorization, appearing in several practical applications, for example, in image processing, computer vision, and signal processing. Let $F : \mathbb{R}^n \longrightarrow \mathbb{R}^n$ be an operator defined by*

$$F(x) = Kx + q, \quad K = AA^\top + B + C, q \in \mathbb{R}^n, \tag{31}$$

*where $A \in \mathbb{R}^{n \times n}$ is a matrix, $B \in \mathbb{R}^{n \times n}$ is a skew-symmetric matrix ($A$ and $B$ are randomly generated from a normal (Gaussian) distribution with mean equals $0$ and scale equals $0.01$) and $C \in \mathbb{R}^{n \times n}$ is a diagonal matrix with non-negative diagonal entries (randomly generated from the continuous uniform distribution over the interval $[0, 1)$). Therefore, it follows that $K$ is positive semidefinite. The operator $F$ is monotone and bounded in the unit ball with constant $L_F = \|K\|_2 + \|q\|_2$. For $q = \mathbf{0} \in \mathbb{R}^n$, the solution of problem equation 7, is $x^* = \mathbf{0} \in \mathbb{R}^n$.*

The results for Example 5.3, are presented in Fig. 2. From this figure, we can see that Algorithm 1 always works better than MPM for any $m \geq 1$.

## 6 CONCLUSION

In this paper, we studied two classes of variational inequality problems. The first is classical constrained (i.e., without functional constraints) variational inequality and the second is the same problem with functional constrained (inequality type constraints). To solve such problems, we proposed mirror descent-type methods with a weighting scheme for the generated points in each iteration of the algorithms. For the second class of problems, we proposed a mirror descent method by switching between adaptive and non-adaptive steps. We analyzed the proposed methods for the time-varying step sizes and proved the optimal convergence rate of the proposed algorithm concerning the classical variational inequality problems with bounded and $\delta$-monotone operators. We conducted some

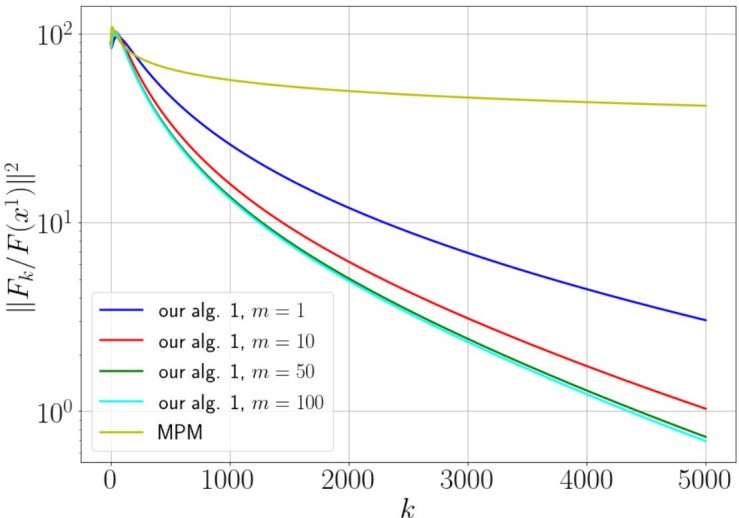

**Figure 2:** Results of Algorithm 1 and Modified Projection Method Khanh & Vuong (2014), for Example 5.3 with $n = 100$.

numerical experiments, which illustrate the advantages of the presented weighting scheme for some examples of the classical variational inequality problem, with a comparison with the modified projection method. As a future work, many directions are connected with the problems under consideration, such as the results for the Lipschitz monotone and strongly monotone operators, also for the stochastic setting of the problem.

**Acknowledgement:** The research is supported by the Ministry of Science and Higher Education of the Russian Federation (Goszadaniye), project No. FSMG-2024-0011.

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

## A  MISSED PROOFS IN SECTION 3

### A.1  PROOF OF THEOREM 3.6

Let $\widetilde{f}(x) := \langle x, F(x^k)\rangle + \frac{1}{\gamma_k}V_\psi(x, x^k)$. From Algorithm 1, we have $x^{k+1} = \arg\min\limits_{x\in Q} \widetilde{f}(x)$. By the optimality condition, we get

$$\left\langle \nabla\widetilde{f}(x^{k+1}), x - x^{k+1}\right\rangle \geq 0, \quad \forall x \in Q.$$

Thus,

$$\left\langle \gamma_k F(x^k) + \nabla\psi(x^{k+1}) - \nabla\psi(x^k), x - x^{k+1}\right\rangle \geq 0, \quad \forall x \in Q.$$

i.e.,

$$\left\langle \gamma_k F(x^k), x^{k+1} - x\right\rangle \leq -\left\langle \nabla\psi(x^k) - \nabla\psi(x^{k+1}), x - x^{k+1}\right\rangle, \quad \forall x \in Q.$$

By Lemma 2.2, for any $x \in Q$, we have

$$-\left\langle \nabla\psi(x^k) - \nabla\psi(x^{k+1}), x - x^{k+1}\right\rangle = -\left(V_\psi(x, x^{k+1}) + V_\psi(x^{k+1}, x^k) - V_\psi(x, x^k)\right)$$
$$= V_\psi(x, x^k) - V_\psi(x, x^{k+1}) - V_\psi(x^{k+1}, x^k).$$

Thus, we get

$$\left\langle \gamma_k F(x^k), x^{k+1} - x\right\rangle \leq V_\psi(x, x^k) - V_\psi(x, x^{k+1}) - V_\psi(x^{k+1}, x^k), \quad \forall x \in Q.$$

This means that for any $x \in Q$, we have

$$\gamma_k \left\langle F(x^k), x^k - x\right\rangle \leq V_\psi(x, x^k) - V_\psi(x, x^{k+1}) - V_\psi(x^{k+1}, x^k) + \left\langle \gamma_k F(x^k), x^k - x^{k+1}\right\rangle.$$

By the Fenchel-Young inequality equation 6, with $\lambda = \sigma > 0$, we find

$$\left\langle \gamma_k F(x^k), x^k - x^{k+1}\right\rangle \leq \frac{\gamma_k^2}{2\sigma}\|F(x^k)\|_*^2 + \frac{\sigma}{2}\|x^k - x^{k+1}\|^2.$$

Therefore, for any $x \in Q$, we get

$$\gamma_k \left\langle F(x^k), x^k - x\right\rangle \leq V_\psi(x, x^k) - V_\psi(x, x^{k+1}) - V_\psi(x^{k+1}, x^k) + \frac{\gamma_k^2}{2\sigma}\|F(x^k)\|_*^2$$
$$+ \frac{\sigma}{2}\|x^k - x^{k+1}\|^2.$$

But from equation 1, we have

$$V_\psi(x^{k+1}, x^k) \geq \frac{\sigma}{2}\|x^{k+1} - x^k\|^2.$$

Thus, for any $x \in Q$, we get the following inequality

$$\left\langle F(x^k), x^k - x\right\rangle \leq \frac{1}{\gamma_k}\left(V_\psi(x, x^k) - V_\psi(x, x^{k+1})\right) + \frac{\gamma_k}{2\sigma}\|F(x^k)\|_*^2. \tag{32}$$

Since $F$ is $\delta$-monotone operator, we get

$$\left\langle F(x), x^k - x\right\rangle - \delta \leq \left\langle F(x^k), x^k - x\right\rangle, \quad \forall x \in Q.$$

Therefore, for any $x \in Q$, we have

$$\langle F(x), x^k - x \rangle \le \frac{1}{\gamma_k} \left( V_\psi(x, x^k) - V_\psi(x, x^{k+1}) \right) + \frac{\gamma_k}{2\sigma} \|F(x^k)\|_*^2 + \delta. \tag{33}$$

By multiplying both sides of equation 33 by $\frac{1}{\gamma_k^m}$, and taking the sum from 1 to $N$, for any $x \in Q$, we get

$$\sum_{k=1}^N \frac{1}{\gamma_k^m} \langle F(x), x^k - x \rangle \le \sum_{k=1}^N \frac{1}{\gamma_k^{m+1}} \left( V_\psi(x, x^k) - V_\psi(x, x^{k+1}) \right) + \frac{1}{2\sigma} \sum_{k=1}^N \frac{\|F(x^k)\|_*^2}{\gamma_k^{m-1}}$$
$$+ \sum_{k=1}^N \frac{\delta}{\gamma_k^m}. \tag{34}$$

But,

$$\sum_{k=1}^N \frac{1}{\gamma_k^m} \langle F(x), x^k - x \rangle = \left( \sum_{k=1}^N \gamma_k^{-m} \right) \left\langle F(x), \frac{1}{\sum_{k=1}^N \gamma_k^{-m}} \sum_{k=1}^N \gamma_k^{-m} x^k - x \right\rangle$$
$$= \left( \sum_{k=1}^N \gamma_k^{-m} \right) \langle F(x), \widehat{x} - x \rangle. \tag{35}$$

For any $x \in Q$, we have

$$\sum_{k=1}^N \frac{1}{\gamma_k^{m+1}} \left( V_\psi(x, x^k) - V_\psi(x, x^{k+1}) \right)$$

$$= \frac{1}{\gamma_1^{m+1}} \left( V_\psi(x, x^1) - V_\psi(x, x^2) \right) + \sum_{k=2}^{N-1} \frac{1}{\gamma_k^{m+1}} \left( V_\psi(x, x^k) - V_\psi(x, x^{k+1}) \right)$$

$$+ \frac{1}{\gamma_N^{m+1}} \left( V_\psi(x, x^N) - V_\psi(x, x^{N+1}) \right)$$

$$\le \frac{1}{\gamma_1^{m+1}} V_\psi(x, x^1) + \frac{1}{\gamma_N^{m+1}} V_\psi(x, x^N) + \sum_{k=2}^{N-1} \frac{1}{\gamma_k^{m+1}} V_\psi(x, x^k) - \frac{1}{\gamma_1^{m+1}} V_\psi(x, x^2)$$

$$- \sum_{k=2}^{N-1} \frac{1}{\gamma_k^{m+1}} V_\psi(x, x^{k+1})$$

$$= \frac{1}{\gamma_1^{m+1}} V_\psi(x, x^1) + \sum_{k=2}^N \frac{1}{\gamma_k^{m+1}} V_\psi(x, x^k) - \sum_{k=2}^N \frac{1}{\gamma_{k-1}^{m+1}} V_\psi(x, x^k)$$

$$= \frac{1}{\gamma_1^{m+1}} V_\psi(x, x^1) + \sum_{k=2}^N \left( \frac{1}{\gamma_k^{m+1}} - \frac{1}{\gamma_{k-1}^{m+1}} \right) V_\psi(x, x^k).$$

Due to equation 15, we find that $V_\psi(x, x^k) \le V_\psi(x, x^1)$, $\forall x \in Q, \forall k = 2, \ldots, N$. Hence we get

$$\sum_{k=1}^N \frac{1}{\gamma_k^{m+1}} \left( V_\psi(x, x^k) - V_\psi(x, x^{k+1}) \right) \le \frac{1}{\gamma_1^{m+1}} V_\psi(x, x^1) + V_\psi(x, x^1) \sum_{k=2}^N \left( \frac{1}{\gamma_k^{m+1}} - \frac{1}{\gamma_{k-1}^{m+1}} \right)$$

$$= \frac{1}{\gamma_1^{m+1}} V_\psi(x, x^1) + V_\psi(x, x^1) \left( -\frac{1}{\gamma_1^{m+1}} + \frac{1}{\gamma_N^{m+1}} \right)$$

$$= \frac{V_\psi(x, x^1)}{\gamma_N^{m+1}}. \tag{36}$$

Therefore, from equation 34, equation 35, and equation 36, we get the following

$$\left( \sum_{k=1}^N \gamma_k^{-m} \right) \max_{x \in Q} \langle F(x), \widehat{x} - x \rangle \le \frac{R^2}{\gamma_N^{m+1}} + \frac{1}{2\sigma} \sum_{k=1}^N \frac{\|F(x^k)\|_*^2}{\gamma_k^{m-1}} + \sum_{k=1}^N \frac{\delta}{\gamma_k^m}.$$

By dividing by $\sum_{k=1}^{N} \gamma_k^{-m}$, we get the desired inequality

$$\max_{x \in Q} \langle F(x), \widehat{x} - x \rangle \leq \frac{1}{\sum_{k=1}^{N} \gamma_k^{-m}} \left( \frac{R^2}{\gamma_N^{m+1}} + \frac{1}{2\sigma} \sum_{k=1}^{N} \frac{\|F(x^k)\|_*^2}{\gamma_k^{m-1}} \right) + \delta.$$

## A.2 PROOF OF COROLLARY 3.7

By setting $m = -1$ in equation 16, we get the following inequality

$$\text{Gap}(\widetilde{x}) \leq \frac{1}{\sum_{k=1}^{N} \gamma_k} \left( R^2 + \frac{1}{2\sigma} \sum_{k=1}^{N} \gamma_k^2 \|F(x^k)\|_*^2 \right) + \delta, \tag{37}$$

where $\widetilde{x} = \frac{1}{\sum_{k=1}^{N} \gamma_k} \sum_{k=1}^{N} \gamma_k x^k$.

**Case 1 (non-adaptive rule).** When $\gamma_k = \frac{\sqrt{2\sigma}}{L_F \sqrt{k}}$, $k = 1, 2, \ldots, N$, and since $\|F(x^k)\|_* \leq L_F$, then by substitution in equation 37 we find

$$\text{Gap}(\widetilde{x}) = \max_{x \in Q} \langle F(x), \widetilde{x} - x \rangle \leq \frac{L_F}{\sqrt{2\sigma}} \cdot \frac{R^2 + \sum_{k=1}^{N} \frac{1}{k}}{\sum_{k=1}^{N} \frac{1}{\sqrt{k}}} + \delta,$$

where $\widetilde{x} = \frac{1}{\sum_{k=1}^{N} \frac{1}{\sqrt{k}}} \sum_{k=1}^{N} \frac{1}{\sqrt{k}} x^k$. But

$$\sum_{k=1}^{N} \frac{1}{k} \leq 1 + \log(N), \quad \text{and} \quad \sum_{k=1}^{N} \frac{1}{\sqrt{k}} \geq 2\sqrt{N+1} - 2, \quad \forall N \geq 1.$$

Therefore,

$$\text{Gap}(\widetilde{x}) \leq \frac{L_F}{\sqrt{2\sigma}} \cdot \frac{R^2 + 1 + \log(N)}{2\sqrt{N+1} - 2} + \delta \leq \frac{L_F}{\sqrt{\sigma}} \cdot \frac{1 + R^2 + \log(N)}{\sqrt{N}} + \delta.$$

Where in the last inequality, we used the fact $2\sqrt{2}\left(\sqrt{N+1} - 1\right) \geq \sqrt{N}, \forall N \geq 1$.

**Case 2 (adaptive rule).** When $\gamma_k = \frac{\sqrt{2\sigma}}{\|F(x^k)\|_* \sqrt{k}}$, $k = 1, 2, \ldots, N$, and since $\|F(x^k)\|_* \leq L_F$, then by substitution in equation 37 we find

$$\text{Gap}(\widetilde{x}) \leq \frac{L_F}{\sqrt{2\sigma}} \cdot \frac{R^2 + \sum_{k=1}^{N} \frac{1}{k}}{\sum_{k=1}^{N} \frac{1}{\sqrt{k}}} + \delta \leq \frac{L_F}{\sqrt{\sigma}} \cdot \frac{1 + R^2 + \log(N)}{\sqrt{N}} + \delta,$$

where $\widetilde{x} = \frac{1}{\sum_{k=1}^{N} \left(\|F(x^k)\|_* \sqrt{k}\right)^{-1}} \sum_{k=1}^{N} \left(\|F(x^k)\|_* \sqrt{k}\right)^{-1} x^k$.

## A.3 PROOF OF COROLLARY 3.8

By setting $m = 0$ in equation 16, we get the following inequality

$$\text{Gap}(\overline{x}) \leq \frac{1}{N} \left( \frac{R^2}{\gamma_N} + \frac{1}{2\sigma} \sum_{k=1}^{N} \gamma_k \|F(x^k)\|_*^2 \right) + \delta, \tag{38}$$

where $\overline{x} = \frac{1}{N} \sum_{k=1}^{N} x^k$.

**Case 1 (non-adaptive rule).** When $\gamma_k = \frac{\sqrt{2\sigma}}{L_F \sqrt{k}}$, $k = 1, 2, \ldots, N$, and since $\|F(x^k)\|_* \leq L_F$, then by substitution in equation 38 we find

$$\text{Gap}(\overline{x}) \leq \frac{1}{N} \left( \frac{R^2 L_F \sqrt{N}}{\sqrt{2\sigma}} + \frac{L_F}{\sqrt{2\sigma}} \sum_{k=1}^{N} \frac{1}{\sqrt{k}} \right) + \delta.$$

But
$$\sum_{k=1}^{N} \frac{1}{\sqrt{k}} \leq 2\sqrt{N}, \quad \forall N \geq 1.$$

Therefore,
$$\text{Gap}(\overline{x}) \leq \frac{1}{N} \cdot \frac{L_F}{\sqrt{2\sigma}} \left(R^2\sqrt{N} + 2\sqrt{N}\right) + \delta = \frac{L_F\left(2 + R^2\right)}{\sqrt{2\sigma}} \cdot \frac{1}{\sqrt{N}} + \delta.$$

**Case 2 (adaptive rule).** When $\gamma_k = \frac{\sqrt{2\sigma}}{\|F(x^k)\|_* \sqrt{k}}$, $k = 1, 2, \ldots, N$, and since $\|F(x^k)\|_* \leq L_F$, then by substitution in equation 38 we find

$$
\begin{aligned}
\text{Gap}(\overline{x}) &\leq \frac{1}{N} \left(\frac{R^2\sqrt{N}\|F(x^N)\|_*}{\sqrt{2\sigma}} + \frac{1}{2\sigma}\sum_{k=1}^{N} \frac{\sqrt{2\sigma}}{\sqrt{k}}\|F(x^k)\|_*\right) + \delta \\
&\leq \frac{1}{N}\left(\frac{R^2 L_F \sqrt{N}}{\sqrt{2\sigma}} + \frac{L_F}{\sqrt{2\sigma}}\sum_{k=1}^{N}\frac{1}{\sqrt{k}}\right) + \delta \\
&\leq \frac{1}{N} \cdot \frac{L_F}{\sqrt{2\sigma}}\left(\sqrt{N}R^2 + 2\sqrt{N}\right) + \delta \\
&= \frac{L_F(2 + R^2)}{\sqrt{2\sigma}} \cdot \frac{1}{\sqrt{N}} + \delta.
\end{aligned}
$$

### A.4   PROOF OF COROLLARY 3.9

Let us see the non-adaptive rule (in a similar way we can consider the adaptive rule). When $\gamma_k = \frac{\sqrt{2\sigma}}{L_F\sqrt{k}}$, $k = 1, 2, \ldots, N$, and since $\|F(x^k)\|_* \leq L_F$, then by substitution in equation 16 we find

$$\text{Gap}(\widehat{x}) \leq \frac{L_F}{\sqrt{2\sigma}} \cdot \frac{1}{\sum_{k=1}^{N}\left(\sqrt{k}\right)^m} \left(R^2\left(\sqrt{N}\right)^{m+1} + \sum_{k=1}^{N}\left(\sqrt{k}\right)^{m-1}\right) + \delta.$$

But, for any $m \geq 1$ and $N \geq 1$,

$$\int_0^N \left(\sqrt{k}\right)^m dk \leq \sum_{k=1}^{N}\left(\sqrt{k}\right)^m \implies \sum_{k=1}^{N}\left(\sqrt{k}\right)^m \geq \frac{2\left(\sqrt{N}\right)^{m+2}}{m+2},$$

and

$$\sum_{k=1}^{N}\left(\sqrt{k}\right)^{m-1} \leq N\left(\sqrt{N}\right)^{m-1} = \left(\sqrt{N}\right)^{m+1}, \quad \forall m \geq 1, N \geq 1.$$

Therefore,

$$
\begin{aligned}
\text{Gap}(\widehat{x}) &\leq \frac{L_F}{\sqrt{2\sigma}} \cdot \frac{m+2}{2\left(\sqrt{N}\right)^{m+2}}\left(R^2\left(\sqrt{N}\right)^{m+1} + \left(\sqrt{N}\right)^{m+1}\right) + \delta \\
&= \frac{L_F(m+2)(1 + R^2)}{2\sqrt{2\sigma}} \cdot \frac{1}{\sqrt{N}} + \delta = O\left(\frac{1}{\sqrt{N}}\right) + \delta.
\end{aligned}
$$

## B   MISSED PROOFS IN SECTION 4

### B.1   PROOF OF THEOREM 4.2

Similar to what was done in the proof of Theorem 3.6, we find that for any $k \in I$ and $x \in Q$ (see equation 32),

$$\langle F(x^k), x^k - x\rangle \leq \frac{1}{\gamma_k^F}\left(V_\psi(x, x^k) - V_\psi(x, x^{k+1})\right) + \frac{\gamma_k^F}{2\sigma}\|F(x^k)\|_*^2.$$

By multiplying both sides of the previous inequality with $\frac{1}{(\gamma_k^F)^m}$, and since $F$ is $\delta$-monotone, i.e.,

$$\left\langle F(x^k), x^k - x \right\rangle \geq \left\langle F(x), x^k - x \right\rangle - \delta, \quad \forall x \in Q,$$

we get (for any $k \in I$ and $x \in Q$)

$$\frac{1}{(\gamma_k^F)^m} \left\langle F(x), x^k - x \right\rangle \leq \frac{1}{(\gamma_k^F)^{m+1}} \left( V_\psi(x, x^k) - V_\psi(x, x^{k+1}) \right) + \frac{\|F(x^k)\|_*^2}{2\sigma(\gamma_k^F)^{m-1}} + \frac{\delta}{(\gamma_k^F)^m}. \tag{39}$$

Also, for any $k \in J$ and $x \in Q$, we have

$$\frac{g(x^k) - g(x)}{(\gamma_k^g)^m} \leq \frac{1}{(\gamma_k^g)^{m+1}} \left( V_\psi(x, x^k) - V_\psi(x, x^{k+1}) \right) + \frac{\|\nabla g(x^k)\|_*^2}{2\sigma(\gamma_k^g)^{m-1}}. \tag{40}$$

By taking the summation, in each side of equation 39 and equation 40, over productive and non-productive steps, with $\gamma_k = \gamma_k^F$ if $k \in I$ and $\gamma_k = \gamma_k^g$ if $k \in J$, we get the following (for any $x \in Q$)

$$\sum_{k \in I} (\gamma_k^F)^{-m} \langle F(x), x^k - x \rangle + \sum_{k \in J} (\gamma_k^g)^{-m} (g(x^k) - g(x))$$

$$\leq \sum_{k=1}^N \frac{1}{\gamma_k^{m+1}} \left( (V_\psi(x, x^k) - V_\psi(x, x^{k+1})) \right) + \frac{1}{2\sigma} \sum_{k \in I} \frac{\|F(x^k)\|_*^2}{(\gamma_k^F)^{m-1}} + \frac{1}{2\sigma} \sum_{k \in J} \frac{\|\nabla g(x^k)\|_*^2}{(\gamma_k^g)^{m-1}}$$

$$+ \sum_{k \in I} \frac{\delta}{(\gamma_k^F)^m}. \tag{41}$$

But for any $x \in Q$, we have

$$\sum_{k \in I} (\gamma_k^F)^{-m} \left\langle F(x), x^k - x \right\rangle = \left\langle F(x), \sum_{k \in I} (\gamma_k^F)^{-m} x^k - \sum_{k \in I} (\gamma_k^F)^{-m} x \right\rangle$$

$$= \left( \sum_{k \in I} (\gamma_k^F)^{-m} \right) \left\langle F(x), \frac{1}{\sum_{k \in I} (\gamma_k^F)^{-m}} \sum_{k \in I} (\gamma_k^F)^{-m} x^k - x \right\rangle$$

$$= \left( \sum_{k \in I} (\gamma_k^F)^{-m} \right) \langle F(x), \widehat{x} - x \rangle.$$

Thus, from the last inequality and equation 41, for any $x \in Q$, we get

$$\left( \sum_{k \in I} (\gamma_k^F)^{-m} \right) \langle F(x), \widehat{x} - x \rangle \leq \sum_{k=1}^N \frac{1}{\gamma_k^{m+1}} \left( (V_\psi(x, x^k) - V_\psi(x, x^{k+1})) \right) + \frac{1}{2\sigma} \sum_{k \in I} \frac{\|F(x^k)\|_*^2}{(\gamma_k^F)^{m-1}}$$

$$+ \frac{1}{2\sigma} \sum_{k \in J} \frac{\|\nabla g(x^k)\|_*^2}{(\gamma_k^g)^{m-1}} + \delta \sum_{k \in I} (\gamma_k^F)^{-m}$$

$$- \sum_{k \in J} (\gamma_k^g)^{-m} (g(x^k) - g(x)). \tag{42}$$

Since, for any $k \in J$, we have

$$g(x^k) - g(x^*) \geq g(x^k) > \varepsilon > 0. \tag{43}$$

Then by the convexity of the function $g$, for any $x \in Q$, we have

$$- \sum_{k \in J} (\gamma_k^g)^{-m} (g(x^k) - g(x))$$

$$= - \sum_{k \in J} (\gamma_k^g)^{-m} (g(x^k) - g(x^*)) + \sum_{k \in J} (\gamma_k^g)^{-m} (g(x) - g(x^*))$$

$$< -\varepsilon \sum_{k \in J} (\gamma_k^g)^{-m} + \sum_{k \in J} (\gamma_k^g)^{-m} \langle \nabla g(x), x - x^* \rangle$$

$$\leq -\varepsilon \sum_{k \in J} (\gamma_k^g)^{-m} + M_g D \sum_{k \in J} (\gamma_k^g)^{-m}, \tag{44}$$

where in the last inequality, we used the Cauchy-Schwartz inequality and the fact that $\|\nabla g(x)\|_* \leq M_g, \forall x \in Q$, and $Q$ is bounded with a diameter $D > 0$.

For any $x \in Q$, we have (see equation 36)

$$\sum_{k=1}^{N} \frac{1}{\gamma_k^{m+1}} \left( V_\psi(x, x^k) - V_\psi(x, x^{k+1}) \right) \leq \frac{V_\psi(x, x^1)}{\gamma_N^{m+1}}. \tag{45}$$

Therefore, by combining equation 44 and equation 45 with equation 42, for any $x \in Q$, we get the following

$$\left( \sum_{k \in I} (\gamma_k^F)^{-m} \right) \langle F(x), \widehat{x} - x \rangle < \frac{V_\psi(x, x^1)}{\gamma_N^{m+1}} + \frac{1}{2\sigma} \sum_{k \in I} \frac{\|F(x^k)\|_*^2}{(\gamma_k^F)^{m-1}} + \frac{1}{2\sigma} \sum_{k \in J} \frac{\|\nabla g(x^k)\|_*^2}{(\gamma_k^g)^{m-1}}$$
$$+ \delta \sum_{k \in I} (\gamma_k^F)^{-m} - (\varepsilon - M_g D) \sum_{k \in J} (\gamma_k^g)^{-m}.$$

By dividing both sides of the last inequality by $\sum_{k \in I} (\gamma_k^F)^{-m} \neq 0$, and taking into account that $\max_{x \in Q} V_\psi(x, x^1) \leq R^2$, we get the desired inequality equation 26.

## C  ANALYSIS OF ALGORITHM 2 WITH TIME-VARYING STEP SIZE RULES

Let us take the following time-varying step size rules

$$\gamma_k = \begin{cases} \gamma_k^F := \frac{\sqrt{2\sigma}}{L_F \sqrt{k}}, & \text{or} \quad \gamma_k^F := \frac{\sqrt{2\sigma}}{\|F(x^k)\|_* \sqrt{k}}; & \text{if } k \in I, \\ \gamma_k^g := \frac{\sqrt{2\sigma}}{M_g \sqrt{k}}, & \text{or} \quad \gamma_k^g := \frac{\sqrt{2\sigma}}{\|\nabla g(x^k)\|_* \sqrt{k}}; & \text{if } k \in J, \end{cases} \tag{46}$$

and first, let us show for Algorithm 2, with equation 46) that $|I| \neq 0$. For this, let us assume that $|I| = 0$, therefore $|J| = N$, i.e., all steps are non-productive.

Let $M := L_F$ when we have a productive step and $M := M_g$ when we have a non-productive step. From equation 43 and equation 46 (we will use the non-adaptive rules, and similarly, we can find the same results by using the adaptive rules) we get

$$\sum_{k=1}^{N} \frac{g(x^k) - g(x^*)}{\gamma_k^m} > \sum_{k=1}^{N} \frac{\varepsilon}{\gamma_k^m} = \frac{\varepsilon M^m}{(\sqrt{2\sigma})^m} \sum_{k=1}^{N} \left( \sqrt{k} \right)^m, \tag{47}$$

and for all $k \in J = \{1, \dots, N\}$, we get

$$\sum_{k=1}^{N} \frac{g(x^k) - g(x^*)}{\gamma_k^m} \leq \frac{R^2}{\gamma_N^{m+1}} + \frac{1}{2\sigma} \sum_{k=1}^{N} \frac{\|\nabla g(x^k)\|_*^2}{\gamma_k^{m-1}}$$
$$\leq \frac{M^{m+1}}{(\sqrt{2\sigma})^{m+1}} \left( R^2 \left( \sqrt{N} \right)^{m+1} + \sum_{k=1}^{N} \left( \sqrt{k} \right)^{m-1} \right).$$

But, it can be verified (numerically) that for a sufficiently big number of iterations $N$ (dependently on suitable values of the parameters $R > 0, m \geq -1, M > 0, \varepsilon > 0, \sigma > 0$), the following inequality holds

$$\frac{M^{m+1}}{(\sqrt{2\sigma})^{m+1}} \left( R^2 \left( \sqrt{N} \right)^{m+1} + \sum_{k=1}^{N} \left( \sqrt{k} \right)^{m-1} \right) < \frac{\varepsilon M^m}{(\sqrt{2\sigma})^m} \sum_{k=1}^{N} \left( \sqrt{k} \right)^m. \tag{48}$$

Therefore, for a sufficiently big number $N \gg 1$, we get

$$\sum_{k=1}^{N} \frac{g(x^k) - g(x^*)}{\gamma_k^m} < \frac{\varepsilon M^m}{(\sqrt{2\sigma})^m} \sum_{k=1}^{N} \left( \sqrt{k} \right)^m.$$

So, we have a contradiction with equation 47. This means that $|I| \neq 0$.

**Remark C.1.** *Note that the reverse inequality of equation 48, i.e.,*

$$\sum_{k=1}^{N} \left(\sqrt{k}\right)^m \leq \frac{M}{\varepsilon\sqrt{2\sigma}} \left( R^2 \left(\sqrt{N}\right)^{m+1} + \sum_{k=1}^{N} \left(\sqrt{k}\right)^{m-1} \right),$$

*for any $m \geq -1, M > 0, R > 0, \sigma > 0$ and $\varepsilon \leq \frac{M}{\sqrt{2\sigma}}$, holds for at least $N = 1$. This means that by choosing $\varepsilon \leq \frac{M}{\sqrt{2\sigma}} (\forall M >, \sigma > 0)$, by Algorithm 2 with equation 46, we have at least one productive step for any $m \geq -1$ and $R > 0$.*

Now let us analyze the convergence of Algorithm 2, by taking the time-varying step size rules equation 46.

Let $M := \max\{L_F, M_g\}$. By using the non-adaptive rules from equation 46 (we also can conclude the same results if we take the adaptive step size rules), and since $\|F(x^k)\|_* \leq L_F \leq M$ and $\|\nabla g(x^k)\|_* \leq M_g \leq M$, then for any $m > 0$, from Theorem 4.2, we have

$$\text{Gap}(\widehat{x}) = \max_{x \in Q} \langle F(x), \widehat{x} - x \rangle$$

$$< \frac{\left(\sqrt{2\sigma}\right)^m}{M^m \sum_{k \in I} \left(\sqrt{k}\right)^m} \left( \frac{R^2 M^{m+1} \left(\sqrt{N}\right)^{m+1}}{\left(\sqrt{2\sigma}\right)^{m+1}} + \frac{1}{2\sigma} \sum_{k=1}^{N} \frac{M^{m+1} \left(\sqrt{k}\right)^{m-1}}{\left(\sqrt{2\sigma}\right)^{m-1}} \right.$$

$$\left. + MD \sum_{k \in J} \frac{M^m \left(\sqrt{k}\right)^m}{\left(\sqrt{2\sigma}\right)^m} \right) + \delta$$

$$= \frac{M}{\sqrt{2\sigma}} \cdot \frac{1}{\sum_{k \in I} \left(\sqrt{k}\right)^m} \left( R^2 \left(\sqrt{N}\right)^{m+1} + \sum_{k=1}^{N} \left(\sqrt{k}\right)^{m-1} + \sqrt{2\sigma}D \sum_{k \in J} \left(\sqrt{k}\right)^m \right) + \delta$$

$$\leq \frac{M}{\sqrt{2\sigma}} \cdot \frac{1}{\sum_{k \in I} \left(\sqrt{k}\right)^m} \left( R^2 \left(\sqrt{N}\right)^{m+1} + N \left(\sqrt{N}\right)^{m-1} + \sqrt{2\sigma}D|J| \left(\sqrt{N}\right)^m \right) + \delta$$

$$= \frac{M(1 + R^2) \left(\sqrt{N}\right)^{m+1} + \sqrt{2\sigma}MD|J| \left(\sqrt{N}\right)^m}{\sqrt{2\sigma} \sum_{k \in I} \left(\sqrt{k}\right)^m} + \delta.$$

Now, by setting

$$\frac{M(1 + R^2) \left(\sqrt{N}\right)^{m+1} + \sqrt{2\sigma}MD|J| \left(\sqrt{N}\right)^m}{\sqrt{2\sigma} \sum_{k \in I} \left(\sqrt{k}\right)^m} \leq \varepsilon,$$

and since $|I| \leq N$, we get

$$\frac{M(1 + R^2) \left(\sqrt{N}\right)^{m+1} + \sqrt{2\sigma}MD|J| \left(\sqrt{N}\right)^m}{\sqrt{2\sigma}N \left(\sqrt{N}\right)^m}$$

$$\leq \frac{M(1 + R^2) \left(\sqrt{N}\right)^{m+1} + \sqrt{2\sigma}MD|J| \left(\sqrt{N}\right)^m}{\sqrt{2\sigma} \sum_{k \in I} \left(\sqrt{k}\right)^m} \leq \varepsilon.$$

Thus,

$$\frac{M(1 + R^2)}{\sqrt{2\sigma}\sqrt{N}} + \frac{MD|J|}{N} \leq \varepsilon.$$

Hence, we can formulate the following result.

**Corollary C.2.** *Let $F : Q \longrightarrow E^*$ be a continuous, bounded, and $\delta$-monotone operator. Let $g(x) = \max_{1 \leq i \leq p}\{g_i(x)\}$ be an $M_g$-Lipschitz convex function, where $g_i : Q \longrightarrow \mathbb{R}, \; \forall i = 1, 2, \ldots, p$ are $M_{g_i}$-Lipschitz, and $M_g = \max_{1 \leq i \leq p}\{M_{g_i}\}$. Then for problem, after $N \geq 1$ iterations of Algorithm 2, such that*

$$\frac{M(1 + R^2)}{\sqrt{2\sigma}\sqrt{N}} + \frac{MD|J|}{N} \leq \varepsilon, \tag{49}$$

*for any fixed $m > 0$, with step size rules given in equation 46, it satisfies*

$$\mathrm{Gap}(\widehat{x}) = \max_{x \in Q} \langle F(x), \widehat{x} - x \rangle < \varepsilon + \delta, \quad \text{and} \quad g(\widehat{x}) \leq \varepsilon,$$

*where $\widehat{x} = \frac{1}{\sum_{k \in I}(\gamma_k^f)^{-m}} \sum_{k \in I} \left(\gamma_k^f\right)^{-m} x^k$.*

Now, by setting $m = 0$ in equation 26, with $\overline{x} = \frac{1}{|I|} \sum_{k \in I} x^k$, we get

$$\mathrm{Gap}(\overline{x}) = \max_{x \in Q} \langle F(x), \overline{x} - x \rangle$$

$$< \frac{1}{|I|} \left( \frac{R^2}{\gamma_N} + \sum_{k \in I} \frac{\|F(x^k)\|_*^2}{2\sigma}\gamma_k^F + \sum_{k \in J} \frac{\|\nabla g(x^k)\|_*^2}{2\sigma}\gamma_k^g - (\varepsilon - M_g D)|J| \right) + \delta$$

$$\leq \frac{1}{|I|} \left( \frac{MR^2\sqrt{N}}{\sqrt{2\sigma}} + \frac{M}{\sqrt{2\sigma}}\sum_{k \in I}\frac{1}{\sqrt{k}} + \frac{M}{\sqrt{2\sigma}}\sum_{k \in J}\frac{1}{\sqrt{k}} + MD|J| \right) + \delta$$

$$= \frac{M}{|I|\sqrt{2\sigma}} \left( R^2\sqrt{N} + \sum_{k=1}^{N}\frac{1}{\sqrt{k}} \right) + \frac{MD|J|}{|I|} + \delta$$

$$\leq \frac{M\sqrt{N}}{|I|\sqrt{2\sigma}} \left(2 + R^2\right) + \frac{MD|J|}{|I|} + \delta.$$

Thus, by setting $\frac{M\sqrt{N}}{|I|\sqrt{2\sigma}}\left(2 + R^2\right) + \frac{MD|J|}{|I|} \leq \varepsilon$ and since $|I| \leq N$, we get

$$\frac{M\left(2 + R^2\right)}{\sqrt{2\sigma}\sqrt{N}} + \frac{MD|J|}{N} \leq \varepsilon.$$

Hence, for $m = 0$, we can formulate the following result.

**Corollary C.3.** *Let $F : Q \longrightarrow E^*$ be a continuous, bounded, and $\delta$-monotone operator. Let $g(x) = \max_{1 \leq i \leq p}\{g_i(x)\}$ be an $M_g$-Lipschitz convex function, where $g_i : Q \longrightarrow \mathbb{R}, \; \forall i = 1, 2, \ldots, p$ are $M_{g_i}$-Lipschitz, and $M_g = \max_{1 \leq i \leq p}\{M_{g_i}\}$. Then, after $N \geq 1$ iterations of Algorithm 2, such that*

$$\frac{M\left(2 + R^2\right)}{\sqrt{2\sigma}\sqrt{N}} + \frac{MD|J|}{N} \leq \varepsilon,$$

*with $m = 0$ and step size rules given in equation 46, it satisfies*

$$\mathrm{Gap}(\overline{x}) = \max_{x \in Q} \langle F(x), \overline{x} - x \rangle < \varepsilon + \delta, \quad \text{and} \quad g(\overline{x}) \leq \varepsilon,$$

*where $\overline{x} = \frac{1}{|I|} \sum_{k \in I} x^k$.*

**Remark C.4.** *By setting $m = -1$ in equation 26, with $\widetilde{x} = \frac{1}{\sum_{k \in I}\frac{1}{\sqrt{k}}} \sum_{k \in I} \frac{1}{\sqrt{k}}x^k$, we have*

$$\mathrm{Gap}(\widetilde{x}) < \frac{M}{\sqrt{2\sigma}} \cdot \frac{1}{\sum_{k \in I}\sqrt{k}} \cdot \left( R^2 + \sum_{k=1}^{N}\frac{1}{k} + D\sqrt{2\sigma}\sum_{k \in J}\frac{1}{\sqrt{k}} \right) + \delta$$

$$\leq \frac{M}{\sqrt{2\sigma}} \cdot \frac{1}{2\sqrt{|I| + 1} - 2} \left( R^2 + 1 + \log(N) + D\sqrt{2\sigma}\sum_{k \in J}\frac{1}{\sqrt{k}} \right) + \delta$$

$$\leq \frac{M}{\sqrt{\sigma}\sqrt{|I|}} \left( R^2 + 1 + \log(N) + D\sqrt{2\sigma}\sum_{k \in J}\frac{1}{\sqrt{k}} \right) + \delta.$$

*Thus, after $N \geq 1$ iterations of Algorithm 2 (with $m = -1$), such that*

$$\frac{M}{\sqrt{\sigma}\sqrt{|I|}} \left( R^2 + 1 + \log(N) + D\sqrt{2\sigma} \sum_{k \in J} \frac{1}{\sqrt{k}} \right) \leq \varepsilon,$$

*it satisfies,*

$$\mathrm{Gap}(\widetilde{x}) = \max_{x \in Q} \langle F(x), \overline{x} - x \rangle < \varepsilon + \delta, \quad and \quad g(\widetilde{x}) \leq \varepsilon.$$

