# OpenReview forum: "Mirror Descent Methods with Weighting Scheme for Outputs for Constrained Variational Inequality Problems"
_mathai.club/MathAI/2025/Conference — MathAI 2025 Oral_

### Official Review · Reviewer_b51d · 2025-02-25
**Review for Mirror Descent Methods with Weighting Scheme for Outputs for Constrained Variational Inequality Problems**

**Rating:** 9
**Confidence:** 3

**Review:**

Summary: This paper presents a comprehensive and structured approach to addressing constrained variational inequality problems using mirror descent methods with proposed new weighting schemes. Both algorithms for classical and functional constraints were introduced and analyzed for their convergence properties, providing optimal rates for various scenarios.

The theoretical contributions are strong, supported by solid proofs and detailed derivations. The numerical experiments are thoughtfully designed and clearly show the improvements in the proposed methods compared to existing techniques.

While the mathematical rigor is impressive, a minor suggestion is that the paper could benefit from more intuitive explanations of the key concepts. This would make the work more accessible to readers who may not have a deep background in variational inequalities.

It is also interesting that the baseline compared is the Modified Projection Method from 2014. I wonder whether there have been more developments in variational inequality methods since then.

---

### Official Review · Reviewer_PRSG · 2025-02-26
**Review of Paper 19**

**Rating:** 7
**Confidence:** 3

**Review:**

**Overall Comment**

Authors in this paper propose novel methods for addressing constrained variational inequality problems, particularly those with functional constraints. The introduction of mirror descent method involving switching between adaptive and non-adaptive steps for variational inequality problems with functional constraints makes this work significant in the field of optimization and machine learning.

**Strength**

•	The paper supports its theoretical claims with extensive numerical experiments.

•	The paper's method of switching between adaptive and non-adaptive steps is a significant achievement for solving constrained variational inequality problems.

**Weakness**

•	The text of the paper is dense, heavily laden with mathematical formulas. It can be improved with a better structure. Breaking up long paragraphs, offering more intuitive explanations, adding more visualizations and summarizing key points will make it more readable to the larger audience.

---

### Official Review · Reviewer_cFu6 · 2025-02-27
**A good paper**

**Rating:** 8
**Confidence:** 4

**Review:**

**Contribution of the paper**

The paper considers a general constrained variational inequality problem and its functionally constrained version, and proposes their solution based on mirror descent method (algorithms 1 and 2). The main theoretical results (Theorems 3.5, 4.2) give upper bounds on the error of the weighted step-$k$ iterate, ensuring the convergence of the algorithms. In the experimental part the proposed algorithms are tested on three analytic problems, showing much better performance than the baseline Modified Projection Method.

**Strengths**

The paper is clearly written, the exposition is logical and well-structured. The introduction includes an abundant literature review. The paper includes both theoretical and experimental results. The theoretical results are formulated as rigorous theorems. I have checked the proof of Theorem 3.5 (one of the two main theorems), and it looks correct. The experiments show a significant advantage of the proposed method compared to the Modified Projection Method baseline.

**Weaknesses**

I have several critical comments, addressing which, I believe, could strengthen the paper.

1. The algorithms 1 and 2 are not elementary, in the sense that they reduce solving one optimization problem (the variational inequality problem (7) or (22)) to a sequence of solutions of auxiliary minimization problems. The solution of these auxiliary problems does not seem to be discussed in the paper, so the total complexity of the proposed algorithms in elementary terms is not very clear. I understand that in the special case of quadratic Bregman divergence and simple domains $Q$ the auxiliary problems are simple, and the original variational problems is more complex in general. However, at least two of the three mentioned general special case of VI - minimization problem (Example 3.1) and fixed point problem (example 3.3) - also admit formulation as a minimization problem on $Q$ and resemble the auxiliary problem in its general form in terms of complexity. In these two cases, can we really expect the proposed algorithm to be competitive compared to standard methods, such as basic gradient descent? Also, is the amount of computation per step in the proposed method roughly the same as in the reference MPM method? I suggest to discuss these issues in the paper.

2. I don't see a strong connection between the theoretical and experimental results of the paper. The main theoretical results seem to be the specific quantitative convergence rates of the proposed algorithms, in particular the $\widetilde O(1/\sqrt{n})$ rates of corollaries 3.6-3.8. It is hard to see from figures 1-2 if the experiments confirm these rates or not. The figures use the linear scale for iterations and the logarithmic scale for the error. Such axes are natural for exponential dependencies, but not power-laws as in $\widetilde O(1/\sqrt{n})$.

3. On the other hand, we clearly observe in the figures some phenomena that are not explained by the developed theory. We definitely see that the algorithms with larger $m$ are uniformly better. In particular, the best algorithm corresponds to $m=\infty$, meaning simply taking the last iterate as the output $\widehat x$ instead of weighting with the previous iterates with coefficients $\gamma_k^{-m}$ as proposed in the theoretical part. This raises the natural question why we need this weighting at all - this is not explained by the theoretical results. Incidentally, the three consecutive corollaries 3.6-3.8 look odd - they are almost identical and it's unclear why state all of them.

4. All the three test problems considered in the experiments look highly artificial. It is puzzling why in section 3 the paper emphasizes the extensiveness of the VI problem, and then tests its algorihm only on three obscure examples that no-one really knows or cares about.

---

### Decision · Program_Chairs · 2025-03-08

**Decision:**

Accept (Oral)

**Comment:**

Your article has been accepted and you can give a talk on the article. All articles will be sorted by rating and within the available conference places one author from each article will be invited. If there are not enough places, then you will either have the opportunity to speak remotely or come at your own expense!